# Intersectional stigma and adherence to antiretroviral (ART) among sexual minority men (SMM) living with HIV in Nigeria: A Qualitative Inquiry

Adedotun Ogunbajo[1]*, Oluwatimilehin Sotubo[1], Kehinde Okanlawon[2], Elizabeth Shoyemi[3], Olakunle Oginni[4], Kenneth H. Mayer[5,6]

1 RAND Corporation, Los Angeles, California, United States of America, 2 Social Inclusion, Justice and Empowerment Initiative, Minna, Nigeria, 3 Centre For Population Health Initiatives, Lagos, Nigeria, 4 The Wolfson Centre for Young People's Mental Health, Division of Psychological Medicine and Clinical Neuroscience, Cardiff University, Cardiff, United Kingdom, 5 The Fenway Institute, Fenway Health, Boston, Massachusetts, United States of America, 6 Division of Infectious Diseases, Beth Israel Deaconess Medical Center, Boston, Massachusetts, United States of America

* dotunogunbajo@gmail.com

## Abstract

### Background

Sexual minority men (SMM) living with HIV in Nigeria experience suboptimal outcomes on the HIV care continuum. Experiences of intersectional stigma has been linked to lower likelihood of engagement in HIV services, but little is known about the effects of intersectional stigma on HIV care outcomes for SMM in Nigeria. The current study explored experiences of intersectional stigma among SMM living with HIV in Lagos, Nigeria and its' impact on engagement in HIV care.

### Methods

Between January and February 2022, we conducted semi-structured individual interviews with 30 SMM living with HIV in Lagos, Nigeria and 38 healthcare providers that provide HIV prevention and care services to SMM.

### Results

We found that while participants described experiencing both HIV and sexual minority stigma, they believed sexual minority identity stigma to be more common than HIV-related stigma. Additionally, most participants believed they experience intersectional stigma, especially enacted and anticipated stigma, from their HIV status and sexual minority identity. Respondents utilized various strategies, both negative and positive, to cope with stigma including substance use, identity concealment, social isolation, support systems, and engaging in various hobbies as a form of self-care.

**Data availability statement:** Data requests can be sent to Elizabeth Shoyemi (eshoyemi@cphinigeria.org), the Executive Director of the collaborative non-profit.

**Funding:** This project was made possible with help from the Harvard University Center for AIDS Research (CFAR), an NIH funded program (P30 AI060354), which is supported by the following NIH Co-Funding and Participating Institutes and Centers: NIAID, NCI, NICHD, NIDCR, NHLBI, NIDA, NIMH, NIA, NIDDK, NINR, NIMHD, FIC, and OAR. The content is solely the responsibility of the authors and does not necessarily represent the official views of the National Institutes of Health.

**Competing interests:** The authors have declared that no competing interests exist.

Lastly, while most healthcare providers believe that stigma negatively influenced SMM ability to fully engage in HIV care, most SMM believed stigmatization had minimal to no effect on their engagement in HIV care.

## Discussion

Our findings underscore the importance of developing community (SMM specifically) informed, evidence-based health interventions aimed at reducing the effect of stigma on engagement in HIV care among Nigerian SMM living with HIV, to improve their overall health and quality of life.

## Introduction

Nigeria has the second-highest prevalence of human immunodeficiency virus (HIV) in the world, with over 2 million people living with HIV and an adult HIV prevalence of 1.4% [1]. Sexual minority men (SMM; including men who identify as gay, bisexual, and other sexual minority men) in Nigeria are at high risk for HIV infection. HIV prevalence among SMM in Nigeria has increased significantly, from 14% in 2007 to 17% in 2010 and 23% in 2014 [2]. Factors associated with heightened risk for HIV infection among SMM in Nigeria include being older, having a history of condomless anal sex with another male, and having a history of other sexually transmitted infections (STIs), and engagement in transactional sex [2–5]. Additionally, compared to the general population, SMM living with HIV in Nigeria experience suboptimal outcomes on the HIV care continuum (diagnosis of HIV infection, linkage to HIV care, receipt of HIV care, retention in HIV care, and achievement and maintenance of HIV viral suppression). A systematic and meta-analysis that explored outcomes across the HIV care continuum among SMM across Africa found low rates of HIV status awareness (19%), current use of antiretroviral therapy (ART) (24%), and HIV viral load suppression (25%), which are lower than those of the general population [6]. Among the general population, HIV status awareness was around 39%, ART uptake was 66% and HIV viral load suppression was closer to 72% [7]. Specifically in Nigeria, a study of SMM living with HIV found that only 31% had initiated ART, but of those who had initiated ART, 80% had achieved viral load suppression [8]. Consequently, it is crucial to elucidate barriers to engagement in ART, especially among SMM living with HIV in Nigeria, to design multilevel interventions to address them.

Stigma refers to the structural and social conditions and systems that lead to social exclusion [9]. SMM can hold several minoritized identities based on race, sexual orientation, gender identity, HIV status, socioeconomic status, etc., that can increase their chances of experiencing stigma and discrimination. The general public's perceptions of and attitudes towards sexual minority communities remain largely unaccepting, especially within the context of the illegality of homosexuality in Nigeria [10,11]. Furthermore, intersectional sigma, which is rooted in the legacy of intersectionality scholarship [12], examines the experiences of discrimination and prejudice resulting from inhabiting multiple stigmatized identities [13]. Intersectional stigma

typically occurs across the various domains of the socioecological framework (individual, interpersonal, community, structural) and is premised on the co-occurrence and intersection of multiple identities and conditions [14–16]. Experiences of intersectional stigma have been linked to a lower likelihood of engagement in HIV prevention and care services among SMM [17–19]. However, little is known about the effects of intersectional stigma on HIV care outcomes for individuals with multiple stigmatized identities, especially in high-stigma environments such as Nigeria.

The Conceptual Framework for HIV-Related Stigma, Engagement in Care, and Health Outcomes outlines how intersectional stigma (structural, sexual orientation, and HIV related) predicts poorer HIV care outcomes and their potential mechanisms of action [20]. Guided by this framework, the current study explored experiences of intersectional stigma among SMM living with HIV in Lagos, Nigeria, and its impact on engagement in HIV care. Gaining a better understanding of this phenomenon is pivotal to developing, testing, and scaling effective interventions aimed at reducing stigma and improving HIV prevention and care outcomes for SMM in Nigeria, specifically, and more broadly across sub-Saharan Africa.

## Methods

### Study setting

Between January and February 2022, we conducted semi-structured individuals interviews with 30 SMM living with HIV in Lagos, Nigeria and 38 healthcare providers (HCPs) that provide HIV prevention and care services to SMM. We included perspectives from HCPs because future interventions aimed at improving HIV outcomes for SMM in Nigeria will likely involve them, reinforcing the importance of including them in the current research project. The following inclusion criteria applied to SMM: (1) biologically assigned male sex at birth, (2) 18 years or older; (3) current residence in Lagos, Nigeria, (4) has engaged in oral or anal sex with at least one birth-assigned male partner in the previous year, and (5) lab-confirmed HIV positive diagnosis. Additionally, transgender women were excluded from participation due to differences in lived experiences compared to SMM. Specifically, transgender women have experiences related to their gender expression, presentation, and conformity (or lack thereof) to societal expectations of gender performance that cisgender SMM don't experience. Additionally, renowned researchers within the field of HIV prevention and treatment have called for the need to study transgender populations, separate from SMM, especially within the African context [21–23]. The following inclusion criteria applied to healthcare workers: (1) 18 years or older, (2) current professional certification, and (3) at least 6 months' experience providing services to SMM in Nigeria. SMM were recruited (between December 2021 and February 2022) through a local community-based organization (CBO) that provides sexual health services to SMM in Lagos, Nigeria. Outreach workers at the CBO shared information about the study with the target population during programming events and provided study contact information to individuals who expressed interest. HCPs included staff at the partner CBO and other health organizations that serve SMM in Lagos, Nigeria.

### Procedures

All study procedures were approved by the institutional review boards at Harvard University (Protocol #: IRB21–0914). Written informed consent was obtained from all participants before completing any study-related activities. We assigned each participant a unique four-digit identifier and did not collect any identifying information to maintain confidentiality. Individual interviews were conducted in a private office at the partner CBO. Interviews were conducted entirely in English and digitally recorded. Each interview lasted about 1.5 hours. The interviews explored the following topics: cultural context and lived experiences of SMM in Nigeria; experiences with various types of stigmas (enacted, structural, community, anticipated, and intersectional); coping mechanisms; and engagement in HIV care. We also collected demographic data from all participants. Upon completion of the interview, participants were compensated in Nigerian naira, equivalent to 20 U.S. dollars ($20).

### Data analysis

We utilized a thematic data analysis approach for the current study. A thematic analysis identifies, analyzes, and reports themes found within a qualitative dataset [24]. The interview transcripts were compiled in Microsoft Excel 2024, and a

comprehensive data analysis was conducted in NVivo 14. A member of the study team read the interview transcripts and generated codes based on pertinent interviewee responses to the questions. The codes for each question of interest were then analyzed and categorized into themes. The analysis was inductive (data-driven), with codes and themes generated from the data, rather than deductive (or driven by a theoretical framework). After initial coding was completed, 15% of randomly selected transcripts were double-coded by a second senior study team member. The resulting interrater reliability was 88%.

## Results

### HIV stigma

Experiences of HIV-related stigma described by participants primarily fell into two stigma categories: anticipated and enacted. Participants more commonly reported anticipated stigma than enacted stigma. Participants attributed this to an individual's HIV status not being easily observable and the proliferation of sensitization efforts to reduce HIV-related stigma in the general population.

**Anticipated stigma (expectations of experiencing stigmatizing events)**: Participants described expecting discrimination from others because of their HIV status. Specifically, participants expected to be discriminated against if their HIV status became known to others. This was due to participant uncertainty around the type of response knowledge of their HIV status would elicit. As a result, many participants hid their HIV status, ARTs, and information about their HIV care from family, friends, and society at large, including other members of the SMM community:

> "I hide my HIV status because the stigma is still there…. Some people will say, 'Please don't come and give me HIV.' So, I hide my HIV status, so that people will not stigmatize me."

An HCP echoed these sentiments by sharing how their SMM clients, at the point of starting HIV care, hide their HIV status, including from family members:

> "Most of them don't want their family members to know. At the point of enrollment, they will tell you, we don't want anybody to disclose our status to anybody, and they will even tell you that 'when my appointment is due, I'll come to the facility to come and pick up my drugs, nobody should call me.'"

**Enacted stigma (direct experiences of prejudice, discrimination, and stereotyping based on a specific attribute):** Respondents shared experiences of stigma due to their HIV status. An HCP narrated the experience of a former client:

> "I had a client who tested positive for HIV, and the client started taking his HIV drugs. His roommate asked about the new medicine he was taking. He told his roommate that it's an immune booster drug, but the roommate did not believe, and googled the name of the drug. He found out that he was living with HIV and began stigmatizing him. When he went back to the house, he said he observed that the roommate had changed. The roommate had separated his items including clothes and cutlery from his own. He also said everybody was now cold, that he felt that his roommate had told people in the hostel."

### Sexual minority stigma

Participants described varied experiences of stigma (anticipated, community, enacted, and structural) related to their known or perceived sexual minority status.

**Anticipated stigma**: Participants anticipated discrimination from others due to their sexual orientation, which affected their public presentation and relationships with others. Anticipated stigma resulted in participants concealing their sexual orientation by hiding aspects of their identity to reduce stigmatizing experiences:

> "Almost every day, and anywhere I go, I don't tell people that I'm a gay because I know that I'm not accepted."

An HCP highlighted cases where clients might be inclined to hide their sexual orientation from healthcare staff:

> "They really don't want you to know their sexual preference because they don't know how you're going to feel about it. I tell them, 'don't worry, we are one. Just be plain to me so that I know how to provide service.'"

**Community stigma (experiences within physical neighborhoods)**: Participants described enduring stigma in their residential neighborhoods. An HCP shared the story of a client who was a hairdresser and experienced stigma due to his perceived sexual orientation:

> "I know one client who is gay and has issues in his community. He is into hairdressing, and that was shattered because in the community where he stays, he is stigmatized for his sexual orientation. Now, he only makes hair for a few young ladies, and it will be inside his house. He could not even explore his career; he was limited because of the stigma he would experience in the outside world."

**Enacted stigma**: Beyond experiences of stigma in their neighborhoods, participants described experiences of stigmatization when their sexual orientation was known or suspected and how that sometimes led to extortion and physical violence. One respondent shared their experience of being extorted by a soldier because of their sexual orientation:

> "I logged onto Grindr (social networking app), and I was chatting with this person. He wanted me to come to his house, but I was not comfortable with that, so we agreed to meet at a bar. We chatted on WhatsApp and did a video call, so I felt safe. I went to meet the person but unknown to me, it was a set-up and a soldier walked up to me and grabbed me from the back. I was apprehensive so I begged him, but he said that before he could release me, I had to pay. He withdrew all the money in my bank account, collected my phone and my school ID card."

Another respondent shared the experience of a sexual minority man living with HIV who was also set up and taken to the police station because of his sexual orientation. As a result of the arrest, his family found out about his sexual orientation and HIV status. He recounted that:

> "His dad just threw him out because he's gay and HIV positive. But the mom started crying and was pleading. Later his dad accepted him back but for some months, his dad didn't talk to him."

Experiences of discrimination and unfair treatment within an institutional setting were also mentioned. An HCP described the stigmatizing experience of a client who was previously receiving care at a general hospital:

> "He discovered that whenever he went there to get these [HIV] drugs, the people working in that health center would watch him and make jest of him. Some even started calling him 'Sister Segun.' So, from there, when that thing [stigma] is getting too much, he was like, he can't bear it again. So, he stopped going there."

## HIV stigma versus sexual minority stigma

Both SMM and their HCP believed sexual minority identity stigma to be more common than HIV-related stigma. In addition, HIV is not unique to the SMM community and is now better understood because of general public HIV sensitization campaigns:

> "In Nigeria, the stigma from living with HIV is less compared to the stigma of being a sexual minority man. People are beginning to know that you can use medication for HIV, and it can get to a point where you won't infect someone else. So, the stigma is reduced. But most people still believe being a sexual minority man is a mental disorder."

HIV stigma was believed to emanate more within the SMM community, while sexual minority stigma was thought to be from the larger society:

> "They [SMM] face more stigma from being HIV positive and less from being SMM depending on the community. Within the SMM community, their sexual orientation is not a problem, but their HIV status is a problem. Within society at large, they don't have a problem with you being HIV positive. They have a problem with you being SMM. So, it can vary depending on what you're talking about."

## Intersectional stigma

Participants believed they experienced intersectional stigma, especially enacted and anticipated stigma, from their HIV status and sexual minority identity. Their responses showed that they mainly experienced anticipated stigma. Additionally, some participants expressed some signs of internalized stigma through their linking of their HIV status to their sexual minority identity:

> "It interacts. Because there are times that I used to feel if I'm not a sexual minority man, maybe I wouldn't be living with HIV."

Explaining how both identities can interact, some respondents described the heavy burden of holding multiple stigmatized identities:

> "The[y] do interact because for someone that's only SMM, he's only having to worry about the SMM identity. Someone who is only HIV positive is only having to worry about being HIV positive. Now having the two of them even becomes more difficult as you don't have so many people you can relate with as much as those who are only HIV positive, or those who are only SMM."

> "You know, it's a different thing, if you are gay or you are HIV positive, but being gay and at the same time being HIV positive is more like an overload, something heavy. It's more like you're carrying a lot of burden. It's more like, 'you know what? any moment from now I'm going to drop dead.' So, I mean that's a lot."

Participants also mentioned a few instances where they faced greater stigmatization because of the interaction of their HIV status and sexual orientation. An HCP described the experience of a client at a previous healthcare facility that was made worse with the discovery of his two identities:

> "A client was perceived to be effeminate. The nurses were somewhat irritated and were asking themselves, 'why is he acting like a woman?' but the client was just downcast and wanted to be attended to. His attitude was, 'I came here to seek help, so you people should just attend to me and let me go.' To worsen this guy's situation, they found out he was HIV positive when they ran their tests. So, it became worse and what he got after then was judgment."

Respondents also alluded to the belief that individuals known or perceived to be sexual minority men are automatically believed to be living with HIV:

"I have a friend whose aunt said, 'HIV is a gay's man sickness, it's the homosexual's sickness' she added that 'it's God's punishment to the homosexuals.'"

### Stigma coping strategies

Respondents adopted several strategies, both negative and positive, to cope with experiences of stigma. The most common strategy mentioned was self-acceptance, while their HCPs most often mentioned having a support system. Other coping mechanisms cited by the participants include substance use, identity concealment, social isolation, support systems, and engaging in various hobbies as a form of self-care.

### Substance use

Some participants utilized substances to cope with stigma. Marijuana use, heavy alcohol consumption, and other drug use were common:

"Smoking has really been my companion. When I sit down and start thinking of my life and what my sexuality and HIV status has caused me, it weighs heavy on me. But when I sit down to smoke and drink, I will just be happy."

An HCP reinforced substance use as a common coping strategy for many of his clients:

"I have many of them that are into drugs and it's because they cannot cope with people bombarding them with stigma. They get involved in different types of drugs."

### Identity concealment

Another stigma coping strategy employed by participants was taking the necessary steps to conceal their stigmatized identities. They did not freely disclose their HIV status and sexual minority status because they expected it to lead to discrimination:

"I hide both my identities, like my sexuality and my HIV status. I make sure I hide both because our society today is not accepting."

Respondents also hid their identities by changing mannerisms that may not conform to societal expectations of gender performance:

"It's my walking. I've really worked on hiding it. Now, I tend to walk very masculine so that people will not be able to talk to me anyhow. They might fear me, 'ah! This one can beat somebody o.' Yes, I try to change how I walk, how I behave, and how I talk, and I started growing beard so I can look very manly."

One HCP described the lengths their clients go to hide their identities, including:

"There are certain things they don't say in public, and they have a lot of passwords on their phones. Messages are passworded, everything on their phone is passworded. All this is to avoid stigmatization by keeping people out."

### Social isolation

Participants described avoiding social situations that might involve human interaction as a coping mechanism for stigma:

> "I just don't go out. I can stay indoors. if I have electricity, and I have internet, and I have my subscriptions, I can live through a lot of those things."

While this worked well for some respondents, it did not work so well for others. One of the respondents broke down in tears, explaining how social isolation has been difficult as a coping mechanism:

> "It's not easy. I was just by myself. I don't have any friends, so it has just been me and me alone. [Respondent crying]."

### Self-acceptance

Participants dealt with stigma through self-acceptance and refusal to dwell on their negative experiences:

> "I've made peace with myself. I've read a lot and I've come to terms with my identities. I don't allow all these to bother me anymore. It doesn't really get to me. I live my life as normal and there have been times when I tell people who just got diagnosed with HIV, that I'm also living with HIV. I just live my life as normal as everybody out there, so it doesn't really get to me. It doesn't really weigh me down."

### Support systems

Respondents, especially the HCPs, mentioned that SMM living with HIV relied on support systems to cope with stigmatization. Their support system consisted of peers in the community, healthcare facility, and various support groups:

> "I thank God that we still have friends around us that help us to build confidence. It has not been easy being a SMM living with HIV. Friends have been very helpful. We try to build ourselves and give ourselves courage. We go out to just take ourselves out of thinking and depression. How do we conquer depression? Sometimes we just go out, meet a friend. We gist, laugh, you know, just do those things to keep ourselves together. If not, there'll be a lot of problems. There'll be a lot of suicide."

> "Before, I isolated myself, which caused me a great deal of depression. Then I realized that wasn't helping me. All I just need to do is be myself, be happy, be around people that love me and that care for me. That has really helped me because most times when I feel down, I have places to go to. I just call a friend and ask 'how are you doing? Are you home? I'm coming over.' I'll just go there chill and in no time, I'm not even thinking about my issue anymore."

Various HCPs echoed these sentiments:

> "Some of them [SMM] get integrated into the community and they relate with their peers. I have seen cases where a person moves out of their parents' house to start squatting with another SMM who understands them. So, they largely withdraw from the environment where there is stigmatization."

### Hobbies as self-care

Respondents mentioned using activities such as listening to music, going to the gym, praying, and immersing themselves in work as coping mechanisms:

"The first coping mechanism that I employed back then was listening to music every time with earphones. I was always on earphones every single time to like, not listen to whatever anybody's saying and just be on my own. It helped a lot."

**Impact of stigma on engagement in HIV care**

There were divergent opinions on the impact of stigma on engagement in HIV care. While most HCP believe that stigma negatively influenced SMM ability to engage in HIV care fully, most SMM believed stigmatization had minimal to no effect on their engagement in care. Interestingly, both clients and HCPs noted how stigma might motivate and encourage engagement in HIV care:

"The stigma has even encouraged me to stick to my drugs, to always come to the clinic on time, and to always come for my appointment. The stigma has encouraged me not to stop my medication, rather it has encouraged me to stick to it."

"I'd say the impact has been positive. Everything that happens to a person will either cause a negative or positive response. For someone that knows that the stigma is getting too much, their mindset will be to eradicate the stigma. So, they would be taking the drug to get to that point where the virus becomes undetectable."

Most of the clients claimed that stigmatization had no impact on their engagement with HIV care. They stated that they were determined to achieve better health outcomes, and the determination drove them to adhere to their care:

"I don't think I should allow the stigma to affect me because I know what I am going for. I know what I want. It's your goal that determines how you go about it. So, if I should be stigmatized, I'm hurting myself in that way and I don't want to do that."

This sentiment was echoed by HCPs:

"It doesn't affect their medication, because they know how important it is for them to take their medication. So, they won't want anything to even distract them from even taking their medication at all."

Clients who maintained that stigmatization did not affect their engagement in HIV care mentioned changing their daily routines to conceal their engagement in care and to avoid stigmatization:

"Using meds would raise eyebrows, especially if you're using your medication in front of people who you don't want to know about your HIV status. I use my meds at nights before I sleep because if I use it during the day when everybody is up and about, they could start asking questions and I don't want people to be in my business."

The HCPs and some SMM noted that stigmatization harmed engagement in HIV care. They mentioned that stigmatization sometimes resulted in depression, which might keep SMM from accessing care or taking their medication:

"It has affected me negatively in the past. I was even stigmatizing my own self because I was taking the drugs and I was like, 'how am I even sure this HIV is even real?' Because people were always saying that, 'You are taking drugs that are for HIV people. If you know that you are not HIV positive, then stop taking drugs.' So sometimes I used to have that mindset that I'm not positive, then I'll stop taking drugs. But later, it affected me very much, and then it affected my viral load. So, I had to realize that this is the only way I can fight this virus, not by listening to people's negative talks."

"Sometimes they don't come in. When you call, they'll say, 'I'm not happy, and I'm tired of taking drugs every day.' Then you know this one is already experiencing stigmatization. So, you just must engage in long conversation, 'oh, you need to take this drug. You know how important this drug is, you just must stay alive so that you can fight against anything that is giving you issues.'"

They also mentioned instances of anticipated stigma, which discouraged clients from using their drugs or coming to the healthcare facility when they believed they would meet other people there:

"Because the people around them have not been able to accept them properly, taking their drugs in the presence of their friends has been a very big challenge. Some of them want to go to a hidden place to take their drugs and maybe when they're in a very big place, some of them skip their drugs, which can in turn affect their adherence and the viral load. And then some of them even miss their clinic appointment because they don't want to tell their friends, I want to go for whatever."

## Discussion

This study explored experiences of intersectional stigma among SMM living with HIV in Lagos, Nigeria, and its impact on engagement in HIV care. We found that while participants described experiencing both HIV and sexual minority stigma, they believed sexual minority identity stigma to be more common than HIV-related stigma. Additionally, most participants thought they experienced intersectional stigma, especially enacted and anticipated stigma, from their HIV status and sexual minority identity. Respondents utilized various strategies, both negative and positive, to cope with stigma, including substance use, identity concealment, social isolation, support systems, and engaging in multiple hobbies as a form of self-care. Lastly, while most HCP believe that stigma negatively influenced SMM ability to engage in HIV care fully, most SMM believed stigmatization had minimal to no effect on their engagement in care. These findings provide further evidence elucidating the widespread experiences of stigma among SMM in Nigeria and its potential impact on their engagement in HIV care services.

We found widespread experiences of HIV and sexual minority status stigma among SMM living with HIV in Nigeria. This finding is consistent with previous studies that have found high levels of stigma and discrimination based on HIV status and sexual orientation for SMM in Nigeria [25–29] and across various other West African countries [30–33]. There have been widespread efforts to combat HIV stigma in Nigeria through various public campaigns and sensitization efforts through multiple media outlets, including television, radio, newspapers, billboards, and door-to-door campaigns, which have been demonstrated to increase accepting attitudes towards people living with HIV [34,35]. However, the general public's perceptions of and attitudes towards sexual minority communities remain largely unaccepting, especially within the context of the illegality of homosexuality in Nigeria. A study of heterosexual undergraduate students in Nigeria found that they had a higher tolerance for lesbians than gay/bisexual men, most considered homosexuality a learned behavior and assumed that gay/bisexual men were sexual perverts and sexual abusers, and many supported conversion therapy [36]. Another study of social media users (N = 323) in Nigeria found that 49% disagreed that homosexuals should be considered 'fellow human beings" and 61% disagreed that homosexuals should have equal rights to social services [37]. Additionally, a national survey assessing social perception of LGBT people in Nigeria was conducted in 2022 and found that almost half (48%) believed people were not born with their sexual orientation, and 75% supported the current anti-LGBT laws in Nigeria. Taken together, these findings provide further evidence for how negative attitudes and unacceptance of LGBT individuals in Nigeria have the potential to propagate acts of stigma, discrimination, and even violence, especially for SMM. Consequently, efforts to reduce widespread sexual minority stigma among the general population are needed.

Intersectional stigma—specifically related to sexual minority and HIV status—was commonplace among SMM and echoed by the HCPs. This aligns with previous work that demonstrated that intersectionality theory is a practical framework for understanding and conceptualizing the lived experiences of LGBT communities in Nigeria [38]. Additionally, many of the axes of stigmatization described by our participants map onto a newly proposed conceptual framework on intersectional stigma and HIV continuum outcomes among SMM in sub-Saharan Africa [16]. Specifically, experiences of police brutality, arrests, extortion, discrimination in healthcare settings, discriminatory experiences from various sources (friends and family), psychosocial health, and substance use emerged from our interviews and map onto all levels outlined in the conceptual framework [16]. The concept of intersectional stigma and its application to LGBT health and HIV outcomes is a relatively new and emerging area of scientific inquiry. A recently published scoping review of HIV-related intersectional stigma among sexual and gender minority communities in sub-Saharan Africa found that only 12% of the identified studies utilized an intersectional lens, and the most common intersectional stigmas investigated were HIV, same-sex attraction/behavior, and gender non-conformity stigma [17]. This provides further evidence of the usefulness of intersectionality theory in this work and the need to design, implement, and assess the efficacy of interventions aimed at reducing intersectional stigma, particularly among SMM populations in countries such as Nigeria.

Participants described adopting both positive and negative strategies to cope with experiences of stigma. Some of the negative coping strategies included substance use, identity concealment, and social isolation, which are predisposing factors to poor mental health outcomes, including depressive symptoms, anxiety disorders, post-traumatic stress disorder, and suicidality [27,39–42]. These harmful coping mechanisms are essential to consider in the Nigerian context, especially given the scarcity of mental health professionals and high levels of unmet need for mental health services that has been widely documented in Nigeria [43–45]. Additionally, sexual and gender minority individuals might encounter an additional barrier in finding a mental health provider who has prior training on the unique lived experiences of this community and is willing to take an affirming and nonjudgmental approach to their delivery of mental health services rather than further demonization and utilization of unproven conversion therapy techniques.

Some of the positive coping strategies identified by participants included self-acceptance, social support, and engagement in various hobbies as a practice of self-care. These findings are aligned with a recently published study of SMM in Nigeria, which found that seeking intra-community support (i.e., within the SMM community) and safe spaces and self-acceptance were some coping strategies in response to having experienced sexual minority stigma [42]. Interventions to cultivate positive coping strategies for SMM living with HIV in Nigeria are needed. A systematic review of stigma reduction interventions for SMM living with HIV identified three mechanisms through which stigma can be effectively addressed [46]. These mechanisms were: 1) motivation activation for behavior change on the interpersonal level, including education strategies, 2) knowledge sharing and social empowerment for key stakeholders, such as healthcare provider trainings and peer support, and 3) introspection and self-reflection to propagate structural changes [46]. Informed by this proposed mechanism of change, future work should design stigma reduction interventions that employ an intersectional lens and are culturally relevant to the lived experiences of SMM in Nigeria.

There were mixed findings on the perceived impact of experiences of stigma on engagement in HIV care. While HCPs believed that stigma negatively affected the ability for SMM to be fully engaged in HIV care, most SMM believed stigma had minimal to no effect on their engagement in care. This is in line with a study of SMM living with HIV in Senegal that had mixed findings on the impact of stigma on HIV outcomes: HIV viral load suppression was found to be positively associated with perceived healthcare stigma but negatively associated with enacted stigma [19]. This highlights the complicated mechanisms through which stigma impacts health outcomes.

While there is a dearth in the literature on stigma reduction interventions aimed at improving HIV outcomes, especially in low- and middle-income countries, there are some notable examples. The *Shikamana* intervention study, a randomized controlled trial that aimed to promote ART adherence among SMM living with HIV in Kenya, utilized a peer support approach and found a sixfold increase in odds of HIV viral suppression in the intervention group [47]. An intervention

aimed at integrating emotional well-being and health navigation for SMM living with HIV in Guatemala found a significant increase in HIV viral load suppression [48]. Another randomized clinical efficacy trial of SMM in India aimed at building resilience was efficacious in reducing condomless anal sex acts [49]. While these study findings are promising, they highlight the lack of an intersectional approach to the design and implementation of these interventions, a lack also demonstrated in the published literature [17]. Nigerian SMM living with HIV can encounter stigma on an array of various identity markers, including HIV status, sexual minority identity, socioeconomic class, ethnic group, and gender expression, among others. Consequently, we must design and test interventions that address the various stigmatized identities and societal markers that Nigerian SMM living with HIV occupy. These interventions must be both comprehensive and, to the extent possible, tailored to the unique experiences of intervention participants. It is vital to engage relevant stakeholders in the design and implementation of these interventions to ensure these interventions are acceptable and utilized by the target population. Lastly, randomized controlled trials investigating the impact of stigma-reduction interventions on both HIV prevention and care outcomes among SMM in Nigeria are needed to demonstrate efficacy and effectiveness and to inform the development of programs that can be integrated into care delivery models for SMM in Nigeria.

There were some study limitations. First, study participants were recruited through community-based organizations and in Lagos, Nigeria, thereby limiting the generalizability of sparticipants to underreport experiences of stigma, thereby resulting in a potential underestimation of the pervasiveness of stigma in this group. However, there are several strengths of the study to note, including the inclusion of perspectives from both SMMs and HCPs, which provides a richer, more in-depth understanding of these issues, and the comprehensive inquiry into the type (enacted, structural, community, anticipated, and intersectional) of stigma experienced by participants. There is always a potential that study compensation could influence study findings. Still, given the minimal compensation provided for study participation, we don't believe this is a significant concern in the current study. We also believe in the core tenets of community-based participatory research, which emphasize the need to compensate individuals adequately for their time and efforts. We determined the compensation amount after consulting with community advisory boards and drawing on our understanding of the community we have worked with for over 10 years. We cannot say with certainty that the compensation was not a significant motivator of research participation; no study can state that as a fact. Lastly, we did not review laboratory results and provider medical notes, limiting our ability to explore how intersectional stigma might influence HIV care outcomes beyond information gathered from study participants. Our findings underscore the importance of developing community (SMM specifically) informed, evidence-based health interventions aimed at reducing the effect of stigma on engagement in HIV care among Nigerian SMM living with HIV, to improve the overall health and quality of life for this group.

## Supporting information

**S1 Document. Inclusivity in Global Research.**
(DOCX)

## Acknowledgments

The authors would like to thank the participants and research staff, without whom this study would not have been possible.

## Author contributions

**Conceptualization:** Adedotun Ogunbajo, Kenneth H. Mayer.

**Formal analysis:** Adedotun Ogunbajo, Oluwatimilehin Sotubo.

**Funding acquisition:** Adedotun Ogunbajo, Kenneth H. Mayer.

**Investigation:** Adedotun Ogunbajo, Kehinde Okanlawon, Kenneth H. Mayer.

**Methodology:** Adedotun Ogunbajo, Kenneth H. Mayer.

**Project administration:** Adedotun Ogunbajo, Kehinde Okanlawon, Elizabeth Shoyemi, Kenneth H. Mayer.

**Resources:** Kenneth H. Mayer.

**Software:** Adedotun Ogunbajo.

**Writing – original draft:** Adedotun Ogunbajo.

**Writing – review & editing:** Oluwatimilehin Sotubo, Kehinde Okanlawon, Elizabeth Shoyemi, Olakunle Oginni, Kenneth H. Mayer.

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
