## [Decision Letter · Decision Letter 0]

28 Feb 2025

Dear Dr. Ogunbajo,

Thank you for submitting your manuscript to PLOS ONE. After careful consideration, we feel that it has merit but does not fully meet PLOS ONE’s publication criteria as it currently stands. Therefore, we invite you to submit a revised version of the manuscript that addresses the points raised during the review process.

We look forward to receiving your revised manuscript.

Kind regards,

Adetayo Olorunlana, Ph.D.

Academic Editor

PLOS ONE

Journal Requirements:

Reviewers' comments:

Reviewer's Responses to Questions

**Comments to the Author**

1. Is the manuscript technically sound, and do the data support the conclusions?

Reviewer #1: Yes

2. Has the statistical analysis been performed appropriately and rigorously?

Reviewer #1: I Don't Know

3. Have the authors made all data underlying the findings in their manuscript fully available?

Reviewer #1: Yes

4. Is the manuscript presented in an intelligible fashion and written in standard English?

Reviewer #1: Yes

Reviewer #1: This well-structured and insightful study explores the impact of intersectional stigma on adherence to antiretroviral therapy (ART) among sexual minority men (SMM) living with HIV in Nigeria, using a qualitative approach. The examination of multiple layers of stigma in this context is particularly valuable, as these dimensions are not commonly studied. This research appears highly relevant and opens avenues for future investigations in other settings.

I support the publication of this manuscript with major revisions:

1. Manuscript:

• Please ensure that page numbers and/or line numbers are correctly indicated.

2. Abstract:

• Define the acronym "HCP" to improve clarity for readers unfamiliar with this terminology.

3. Introduction:

• Clarify the distinction between "sexual minority men (SMM)" and "men who have sex with men (MSM)" to ensure conceptual precision.

• In the statement, "Additionally, SMM living with HIV in Nigeria experience suboptimal outcomes on the HIV care continuum," specify whether this comparison is being made against other countries or different population groups.

4. Methods:

• The exclusion of transgender women is justified based on differing lived experiences. However, please provide a more precise explanation of these differences and their implications for the study.

• The manuscript states that Excel was used for data coding. Was no other qualitative data analysis software employed? Additionally, clarify whether coding was cross-validated between researchers to ensure rigor.

5. Results:

• Provide a clear definition of "community" as used in the study.

• Elaborate on the distinction between "community stigma" and "enacted stigma" to enhance conceptual clarity.

• The assertion that "the illegality of homosexual relationships in Nigeria could be the primary reason for this phenomenon" would be more appropriately placed in the Introduction.

• In the statement, "A vast majority of the HCPs and some SMM noted that stigmatization had a negative impact on engagement in HIV care," please specify the number or percentage of participants who expressed this view. In a broader way, specify in the entire results section if several individuals are concerned for a result and define it clearly.

• Clarify whether respondents provided both negative and positive perspectives or whether different individuals contributed distinct viewpoints at different times.

6. Discussion:

• Address the potential impact of participant compensation on study findings.

• Expand the discussion on the implications of the illegality of homosexuality in Nigeria.

• The following sentence is lengthy and somewhat unclear: "It is important to intentionally engage both SMM and relevant stakeholders in the designing and implementation of these interventions and the utilization of a strengths-based and resilience-focused [44], rather than a deficit, approach and incorporation of key tenets of community-based participatory research are paramount to ensuring these interventions are acceptable and utilized by the target population." Please consider rewording for clarity and conciseness.

• Include additional discussion on study limitations to provide a more comprehensive evaluation of its scope and constraints.

**Do you want your identity to be public for this peer review?** For information about this choice, including consent withdrawal, please see our Privacy Policy

Reviewer #1: No

---

## [Decision Letter · Decision Letter 1]

12 Aug 2025

Dear Dr. Ogunbajo,

Thank you for submitting your manuscript to PLOS ONE. After careful consideration, we feel that it has merit but does not fully meet PLOS ONE’s publication criteria as it currently stands. Therefore, we invite you to submit a revised version of the manuscript that addresses the points raised during the review process.

We look forward to receiving your revised manuscript.

Kind regards,

Adetayo Olorunlana, Ph.D.

Academic Editor

PLOS ONE

Journal Requirements:

Reviewers' comments:

Reviewer's Responses to Questions

**Comments to the Author**

Reviewer #1: (No Response)

2. Is the manuscript technically sound, and do the data support the conclusions?

Reviewer #1: Partly

3. Has the statistical analysis been performed appropriately and rigorously?

Reviewer #1: Yes

4. Have the authors made all data underlying the findings in their manuscript fully available?

Reviewer #1: Yes

5. Is the manuscript presented in an intelligible fashion and written in standard English?

Reviewer #1: Yes

Reviewer #1: Please see word file with 2nd round of comments to address.

**Do you want your identity to be public for this peer review?** For information about this choice, including consent withdrawal, please see our Privacy Policy

Reviewer #1: **Yes: ** Tristan Alain

---

## [Decision Letter · Decision Letter 2]

29 Oct 2025

Dear Dr. Ogunbajo,

Thank you for submitting your manuscript to PLOS ONE. After careful consideration, we feel that it has merit but does not fully meet PLOS ONE’s publication criteria as it currently stands. Therefore, we invite you to submit a revised version of the manuscript that addresses the points raised during the review process.

We look forward to receiving your revised manuscript.

Kind regards,

Adetayo Olorunlana, Ph.D.

Academic Editor

PLOS ONE

Journal Requirements:

Reviewers' comments:

Reviewer's Responses to Questions

**Comments to the Author**

Reviewer #1: (No Response)

2. Is the manuscript technically sound, and do the data support the conclusions?

Reviewer #1: Yes

3. Has the statistical analysis been performed appropriately and rigorously?

Reviewer #1: N/A

4. Have the authors made all data underlying the findings in their manuscript fully available?

Reviewer #1: Yes

5. Is the manuscript presented in an intelligible fashion and written in standard English?

Reviewer #1: Yes

Reviewer #1: Please see file attached to adress last comments.

**Do you want your identity to be public for this peer review?** For information about this choice, including consent withdrawal, please see our Privacy Policy

Reviewer #1: No

---

## [Decision Letter · Decision Letter 3]

30 Nov 2025

Intersectional stigma and adherence to antiretroviral (ART) among sexual minority men (SMM) living with HIV in Nigeria: A Qualitative Inquiry

PONE-D-24-43676R3

Dear Dr. Ogunbajo,

We’re pleased to inform you that your manuscript has been judged scientifically suitable for publication and will be formally accepted for publication once it meets all outstanding technical requirements.

Kind regards,

Adetayo Olorunlana, Ph.D.

Academic Editor

PLOS ONE

Additional Editor Comments (optional):

Reviewers' comments:

Reviewer's Responses to Questions

**Comments to the Author**

Reviewer #1: All comments have been addressed

2. Is the manuscript technically sound, and do the data support the conclusions?

Reviewer #1: Partly

3. Has the statistical analysis been performed appropriately and rigorously?

Reviewer #1: N/A

4. Have the authors made all data underlying the findings in their manuscript fully available?

Reviewer #1: Yes

5. Is the manuscript presented in an intelligible fashion and written in standard English?

Reviewer #1: Yes

Reviewer #1: (No Response)

**Do you want your identity to be public for this peer review?** For information about this choice, including consent withdrawal, please see our Privacy Policy

Reviewer #1: No

---

## [Editor Report · Acceptance letter]

PONE-D-24-43676R3

PLOS One

Dear Dr. Ogunbajo,

I'm pleased to inform you that your manuscript has been deemed suitable for publication in PLOS One. Congratulations! Your manuscript is now being handed over to our production team.

Kind regards,

on behalf of

Associate Professor Adetayo Olorunlana

Academic Editor

PLOS One